# Spatiotemporal variability in dengue transmission intensity in Jakarta, Indonesia

**Megan O'Driscoll**[1]\*, **Natsuko Imai**[1], **Neil M. Ferguson**[1], **Sri Rezeki Hadinegoro**[2], **Hindra Irawan Satari**[2], **Clarence C. Tam**[3,4], **Ilaria Dorigatti**[1]

**1** MRC Centre for Global Infectious Disease Analysis, Department of Infectious Disease Epidemiology, Imperial College London, London, United Kingdom, **2** Department of Child Health, Faculty of Medicine Universitas Indonesia, Jakarta, Indonesia, **3** Saw Swee Hock School of Public Health, National University of Singapore and National University Health System, Singapore, **4** London School of Hygiene & Tropical Medicine, London, United Kingdom

\* m.odriscoll@imperial.ac.uk

**Data Availability Statement:** All data used in this analysis are reported in the tables and figures of the manuscript and supporting information or are

## Abstract

### Background

Approximately 70% of the global burden of dengue disease occurs on the Asian continent, where many large urban centres provide optimal environments for sustained endemic transmission and periodic epidemic cycles. Jakarta, the capital of Indonesia, is a densely populated megacity with hyperendemic dengue transmission. Characterization of the spatiotemporal distribution of dengue transmission intensity is of key importance for optimal implementation of novel control and prevention programmes, including vaccination. In this paper we use mathematical models to provide the first detailed description of spatial and temporal variability in dengue transmission intensity in Jakarta.

### Methodology/Principal findings

We applied catalytic models in a Bayesian framework to age-stratified dengue case notification data to estimate dengue force of infection and reporting probabilities in 42 subdistricts of Jakarta. The model was fitted to yearly and average annual data covering a 10-year period between 2008 and 2017. We estimated a long-term average annual transmission intensity of 0.130 (95%CrI: 0.129–0.131) per year in Jakarta province, ranging from 0.090 (95%CrI: 0.077–0.103) to 0.164 (95%CrI: 0.153–0.174) across subdistricts. Annual average transmission intensity in Jakarta province during the 10-year period ranged from 0.012 (95%CrI: 0.011–0.013) in 2017 to 0.124 (95%CrI: 0.121–0.128) in 2016.

### Conclusions/Significance

While the absolute number of dengue case notifications cannot be relied upon as a measure of endemicity, the age-distribution of reported dengue cases provides valuable insights into the underlying nature of transmission. Our estimates from yearly and average annual case notification data represent the first detailed estimates of dengue transmission intensity in Jakarta's subdistricts. These will be important to consider when assessing the population-level impact and cost-effectiveness of potential control and prevention programmes in

publicly available online at http://surveilans-dinkesdki.net/.

**Funding:** We acknowledge grant funding support from the Medical Research Council (Centre grant MR/R015600/1 - NMF, NI, MO'D, ID), the Imperial College Junior Research Fellowship Scheme (ID), the National Institute of General Medical Science MIDAS initiative (grant 5U01GM110721 - NMF) and the Bill and Melinda Gates Foundation (grant OPP1092240 – NMF, NI, MO'D, ID). The funders had no role in study design, data analysis, decision to publish, or preparation of the manuscript.

**Competing interests:** SRH and HIS have been clinical trial and/or study investigators for, and received associated payments from, Sanofi Pasteur, a company involved in the development of dengue vaccines.

Jakarta province, such as the controlled release of *Wolbachia*-carrying mosquitoes and vaccination.

## Author summary

Characterization of the spatiotemporal distribution of dengue transmission intensity, a key measure of population infection risk, can inform the optimal use and deployment of control and prevention programmes. Jakarta, the capital of Indonesia, is a large urban centre with hyperendemic dengue transmission. We fitted catalytic models to age-stratified dengue surveillance data reported in Jakarta's subdistricts from 2008 to 2017. We estimated a long-term average annual transmission intensity of 0.130 (95%CrI: 0.129–0.131) per year in Jakarta province, which varied across subdistricts from 0.090 (95%CrI: 0.077–0.103) per year in Sawah Besar to 0.164 (95%CrI: 0.153–0.174) per year in Pasar Rebo. We observed significant spatiotemporal variation and clustering of transmission intensity in Jakarta. Our estimates obtained from the analysis of yearly and cumulative case-notification data reported between 2008 and 2017 represent the first detailed estimates of average dengue transmission intensity, which will be key to assess the potential impact of future control and prevention programmes in Jakarta province.

## Introduction

Dengue is the most rapidly expanding mosquito-borne viral disease [1]. Increased globalization and rapid urbanization continue to facilitate the geographic expansion of the mosquito vector and virus in tropical regions, where the frequency and magnitude of dengue epidemics has increased dramatically in the past 40 years [2]. Approximately 70% of the global burden of dengue disease occurs on the Asian continent [3], where many large urban centres provide optimal environments for sustained endemic transmission and periodic epidemic cycles.

Indonesia reports the highest number of dengue haemorrhagic fever (DHF) cases in the WHO Southeast Asia Region [1], though experts acknowledge that case numbers are largely under-reported and that reporting practices vary substantially by region [4]. The 1997 World Health Organization (WHO) case definitions are used for dengue reporting in Indonesia, where only dengue haemorrhagic fever (DHF) and dengue shock syndrome (DSS) are notifiable. A cluster-design cross-sectional seroprevalence survey conducted in 2014 estimated an average annual force of infection of 0.140 (95%CI: 0.133–0.147) in urban populations of Indonesia [5]. Jakarta, the capital of Indonesia, is a densely populated megacity with endemic transmission of all 4 dengue serotypes and cyclical epidemics every 3–5 years [6,7]. Current control strategies in Jakarta focus on the removal of mosquito breeding sites by the Mosquito Nest Eradication Program (Pemberantasan Sarang Nyamuk), as well as the periodic use of chemical insecticides for vector management [8].

The characterization of spatial and temporal variability in dengue transmission intensity has become increasingly important in recent years to inform optimal implementation of novel vector control and prevention strategies, including the release of *Wolbachia*-infected mosquitoes and vaccination. The most recent WHO guidelines for use of the world's first dengue vaccine, CYD-TDV (or Dengvaxia), recommend a 'test-before-vaccination' approach with serological testing of individuals prior to vaccination [9], thus presenting significant financial and logistical challenges for the implementation of vaccination campaigns. While the

implementation of individual-based screening and vaccination programmes may not be feasible at the national-level for many countries, geographically targeted interventions implemented at a sub-national level may present a more realistic and cost-effective option in specific settings. Dengue force of infection, which is defined as the per capita rate susceptible individuals acquire infection, is a key measure of transmission intensity and provides valuable insights into the age-related risk of infection and population immunity dynamics. In many dengue endemic countries, routinely collected case notifications are often the only available source of data with which to assess dengue transmission intensity. Though the reliability of dengue case notification data based on clinical diagnoses can vary due to non-specific clinical manifestations caused by dengue and other vector-borne infections, analysis of the relationship between age and disease incidence can provide valuable insights into the underlying nature and intensity of dengue transmission [10].

The age-related patterns of dengue incidence obtained from reported surveillance data can be attributed to the transmission setting and multiple other factors, including the complex immunological interactions of the four dengue virus serotypes, reporting practices, healthcare-seeking behaviour and surveillance capacities. A large proportion of dengue infections result in mild or no symptoms and infection with any one serotype is thought to provide long-term immunity to the same serotype and a short-lived period of cross-protection against infections with heterologous serotypes [11,12]. Severe dengue disease is often, but not always, associated with secondary infections through an immunopathological phenomenon known as antibody-dependent enhancement (ADE), whereby pre-existing anti-DENV antibodies enhance disease severity in secondary heterologous infections [13]. Due to cross-reactivity of dengue antibodies in current serological assays, less is known about tertiary and quaternary infections [14] which are thought to cause less symptomatic disease than primary or secondary infections [15]. For this reason, most mathematical models of dengue transmission assume clinical protection upon secondary infection [16].

Here we apply catalytic models [10] to provide the first detailed description of spatial and temporal variability in dengue transmission intensity in Jakarta, Indonesia. We estimated dengue transmission intensity and reporting probabilities from age-stratified case notification data in 42 subdistricts of Jakarta over a 10-year period between 2008 and 2017. These estimates can inform the implementation of future control and preventative programmes.

## Methods

### Data

In Jakarta province, cases of dengue haemorrhagic fever (DHF) and dengue shock syndrome (DSS) are both notified as DHF to the Ministry of Health in Jakarta. These publicly available data are reported from public and private hospitals as well as *puskesmas* (primary healthcare centres) [17]. Laboratory confirmation of dengue is uncommon, with diagnoses predominantly based on clinical criteria and basic haematology results [18]. Age-stratified case notification data were collated for the 44 subdistricts (administrative level 3) of Jakarta for the years 2008–2017 [17]. DHF cases reported in children <1 year old were excluded from our analyses due to the existence of maternal antibodies which could have potentially predisposed some infants to symptomatic or severe disease [19]. Since the life expectancy in Indonesia is approximately 70 years [20], we assumed that cases reported in the age group ≥75 occurred in individuals aged 75–80. We used population age structure data from the Jakarta Open Data website [21] and from the Ministry of National Development Planning [22]. The population age structure for Jakarta province was available for 2010 and 2014 [22] and the population age structure at the subdistrict-level was available for the years 2014–2016 [21]. From the provincial-level

population age structure for 2010 and 2014, we derived average annual age group-specific rates of population change, which were applied to the 2014 subdistrict-level age structures to obtain approximate subdistrict age structures for the years 2010–2013. In the absence of population data before 2010 and after 2016, we assumed the same population age structure observed in 2010 for the years 2008 and 2009 and the same age structure observed in 2016 for the year 2017 across all subdistricts. The yearly population age structures derived for each subdistrict from the available data are given in Supplementary file (S1 File). The yearly total case numbers and incidence rates of hospitalized DHF for Jakarta province used in this analysis are reported in Table 1 and data by subdistrict are available in Tables A and B in S1 Text. Between 2008 and 2017, over 200,000 hospitalized cases of DHF were reported in Jakarta, which corresponds to an average incidence of 2.07 per 1,000 persons per year. Fig 1 shows the average age-specific incidence of hospitalized DHF averaged over the 10-year period, by region and subdistrict. The Thousand Islands/Kepulauan Seribu regency was omitted from subsequent analyses due to sparse case notification data from this regency, a chain of islands in the Java Sea that are predominantly privately owned.

## Mathematical models

In this analysis we apply two mathematical models. Model 1, which assumes time-constant transmission intensity, was fitted to age-stratified case data averaged over the entire 10-year period to infer long-term average estimates in transmission intensity. This was done at both province and subdistrict levels. Model 2 allows transmission intensity to vary in time and was fitted to yearly data at both province and subdistrict levels. Both models assume transmission of all four dengue virus serotypes [6] and that infection with any one serotype provides lifelong homotypic immunity. The models also assume that transmission intensity is age-constant and, due to lack of serotype-specific data, that the four dengue serotypes were transmitted with equal intensity.

For both Model 1 and Model 2 we explored two model sub-variants, which we denoted model variant S and PS. Under model variant S we assumed that the observed disease incidence (i.e. all hospitalized DHF cases) were caused by secondary infections (i.e. we set $\gamma 1$ to zero). Under model variant PS we assumed that hospitalized DHF cases could be caused by primary or secondary infections, where primary infections were less likely to be symptomatic and reported than secondary infections (i.e. we assumed $\gamma 1 < \rho$).

**Table 1. Total hospitalized dengue haemorrhagic fever (DHF) cases and corresponding incidence rates in Jakarta province reported by hospitals to the Jakarta Ministry of Health Epidemiological surveillance [17].** Data at subdistrict-level are provided in Tables A and B in the S1 Text.

| Year | Hospitalized DHF cases | Incidence of hospitalized DHF cases per 1,000 |
|---|---|---|
| 2008 | 27,656 | 2.980 |
| 2009 | 27,694 | 2.984 |
| 2010 | 28,101 | 3.027 |
| 2011 | 10,552 | 1.115 |
| 2012 | 11,901 | 1.234 |
| 2013 | 18,848 | 1.917 |
| 2014 | 17,916 | 1.789 |
| 2015 | 11,494 | 1.128 |
| 2016 | 38,675 | 3.753 |
| 2017 | 7,870 | 0.764 |
| 2008–2017 | 200,707 | 2.069 |

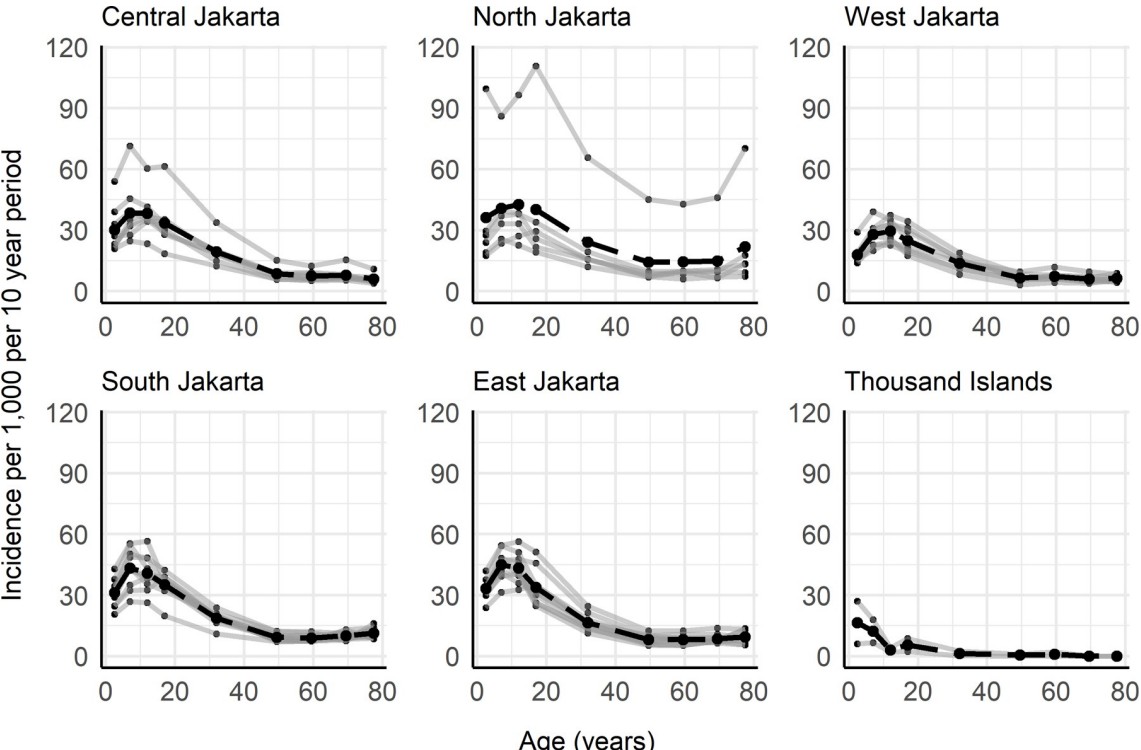

**Fig 1. Cumulative age-specific incidence of hospitalized dengue haemorrhagic fever (DHF) cases by region and subdistrict for the 10-year period between 2008 and 2017 in Jakarta province.** Grey lines show subdistrict age-distributions and black dashed lines show the average age-distribution of hospitalized DHF cases for the region. Dengue case notification data were obtained from the Jakarta Ministry of Health Epidemiological surveillance system (17). Subdistrict incidence rates and population age structures are reported in Tables B and C in the S1 Text.

We defined a multinomial log-likelihood (for details see section 3 of the S1 Text) and used the Markov Chain Monte Carlo (MCMC) Metropolis-Hastings algorithm for parameter inference. We assumed uniform priors on all parameters (range 0–1). Age-specific seroprevalence estimates were obtained using force of infection estimates as detailed in section 3.3 of the S1 Text. The proportion of subdistricts reaching 50%, 70% and 90% seroprevalence at 9 years of age were calculated using average force of infection estimates obtained from the fit of model 1 to average annual age-stratified DHF incidence rates from the 10-year period (2008–2017) in Jakarta's subdistricts.

## Model 1

Model 1 assumes endemic dengue transmission which is constant in time and age. Under these assumptions the incidence of primary, secondary, tertiary and quaternary infections by age can be calculated as described in detail by Imai et al. [10]. The incidence of primary infections in age group *j* is calculated as shown in Eq 1:

$$I_1(j) = \int_{a_j}^{a_{j+1}} 4\lambda(e^{-\lambda a})^4 da \qquad (\text{Eq 1})$$

Equations for the calculation of the incidence of secondary, tertiary and quaternary infections are given in section 2 of the S1 Text. Here λ denotes the force of infection of each

individual serotype, with the total force of infection calculated as four times the serotype-specific force of infection (i.e. 4λ). $a_j$ and $a_{j+1}$ represent the lower and upper bounds of age group $j$. Assuming clinical protection after secondary infection (i.e. that tertiary and quaternary infections are not symptomatic), the average annual incidence of dengue disease per person in age group $j$, $D(j)$, is then calculated as the weighted sum of primary and secondary infection incidence, assuming constant-in-time reporting rates:

$$D(j) = \frac{\rho}{w(j)} \left( I_2(j) + \gamma_1(I_1(j)) + Bw(j) \right) \tag{Eq 2}$$

where ρ denotes the probability that a secondary infection is symptomatic and reported to surveillance, $w(j)$ is the width of age group $j$, $\gamma_1$ is the probability that a primary infection is reported relative to a secondary infection, and B is the probability of reporting non-dengue illnesses as dengue infections (e.g. due to misdiagnosis) relative to a secondary infection. Time-constant basic reproduction numbers ($R_0$) were calculated from serotype-specific force of infection estimates, λ, under two assumptions (equations shown in section 3.4 of the S1 Text). Under assumption 1, we assumed that all infections (primary to quaternary) contribute to onward transmission; under assumption 2, we assumed that only primary and secondary infections are infectious and thus contribute to dengue transmission.

## Model 2

In model 2 we extend model 1 to allow transmission intensity to vary from year to year. In order to estimate the population-level immunity profiles at the beginning of 2008, we assume transmission intensity to have been constant throughout the lifetimes of individuals observed in the first year of our data, i.e. a constant transmission intensity for 80 years prior to and including 2008. The incidence of primary and secondary infections in subsequent years can then be calculated based on the estimated population-level immunity profiles, i.e. the estimated proportion of the population susceptible, S, and the estimated proportion of the population exposed to a monotypic infection, M, by age, as shown in Eqs 3–6.

$$S(a, t_0) = e^{-4\lambda t_0 a} \tag{Eq 3}$$

$$M(a, t_0) = 4e^{-3\lambda t_0 a} \cdot (1 - e^{-\lambda t_0 a}) \tag{Eq 4}$$

$$I_1(a, t) = 4\lambda(t) \cdot S(a, t - 1) \tag{Eq 5}$$

$$I_2(a, t) = 3\lambda(t) \cdot M(a, t - 1) \tag{Eq 6}$$

We fitted the model to the case-notification data reported in each year of the study period (2008–2017) simultaneously, whereby the transmission intensity in each year is dependent on the transmission intensity of all other years. Model 2 was fitted independently to each geographic unit (i.e. at the province- and subdistrict-level, separately), assuming no spatial interactions. The susceptible and monotypic population proportions were estimated by single years of age to account for ageing. These estimates were then averaged across the age groups of our data and disease incidence was calculated as in Eq 2. Due to the presence of maternal immunity potentially increasing the risk of disease in <1-year olds, the model was fitted to case data from 1 to 80 year olds only. Susceptible and monotypic proportions of 1-year olds were added to the population each year at a rate consistent with having been exposed to the previous year's force of infection.

### Spatial analysis

Spatial autocorrelation of dengue force of infection was assessed using the Moran's I-coefficient and local indicator of spatial association (LISA) metrics. The significance of the Moran's I-coefficient was assessed by a Monte-Carlo permutation test with 1,000 simulations, under the null-hypothesis that subdistrict estimates of dengue force of infection are randomly distributed within the province of Jakarta. Subdistrict population density estimates from the 2010 census, available online [23], were used to assess the relationship between subdistrict population density and the estimated dengue force of infection using a simple linear regression model (full details given in section 3.5 of S1 Text).

All analyses were conducted in R statistical software, version 3.4.3 [24].

## Results

All parameter estimates and model fits for Jakarta province, obtained using both model variants *S* (assuming all reported hospitalised DHF cases are due to secondary infections) and *PS* (assuming reported hospitalised DHF cases are due to primary and secondary infections), are given in Table D and S1 Fig of the S1 Text. Model variants *S* and *PS* produced largely similar estimates, particularly when estimates of the probability of primary infections being reported as DHF cases ($\gamma_1$) are negligible. Model variant *PS* was favoured by Deviance Information Criterion (DIC) and therefore for brevity, in the remainder of the manuscript, we focus on the results obtained with model variant *PS*. Estimates from model variant *S* are given in the S1 Text. All force of infection estimates quoted in this manuscript refer to the total force of infection from all dengue serotypes (i.e. $4\lambda$).

An average annual force of infection of 0.130 (95%CrI: 0.129–0.131) was estimated for Jakarta province between 2008 and 2017 using model 1 variant *PS*. This same model produced estimates of the probability of a dengue infection resulting in a reported hospitalized DHF case (reporting rate for secondary infections, $\rho$) of 0.077 (95%CrI: 0.076–0.078), with primary infections being 0.002 (95%CrI: 0.001–0.009) times as likely to result as hospitalized DHF (reporting probability, $\gamma_1$ relative to $\rho$) (Table D in S1 Text). Using model 2, variant PS, the annual average force of infection in Jakarta province varied by year from 0.012 (95%CrI: 0.011–0.013) in 2017 to 0.124 (95%CrI: 0.121–0.128) in 2016 (blue points in Fig 2 and Table D in S1 Text). The range of spatiotemporal variation is shown in Fig 2, where the green points represent the median annual average force of infection estimates obtained at the subdistrict level.

Fig 3 shows the average annual subdistrict force of infection estimates obtained from the fit of model 1 variant *PS* to data averaged over the 10-year period, 2008–2017. We find that Sawah Besar subdistrict had the lowest average annual transmission intensity, with an average force of infection of 0.090 (95%CrI: 0.077–0.103) per year, while Pasar Rebo experienced the highest average force of infection of 0.163 (95%CrI: 0.153–0.174) per year (see Table E in S1 Text for full results). Maps of yearly subdistrict force of infection are shown in S6 Fig in S1 Text. Analysis of spatial autocorrelation of the average annual subdistrict force of infection estimates for the 10-year period using model 1 variant *PS* showed significant spatial clustering of dengue transmission, with a Moran's I-coefficient of 0.29 (p-value = 0.002). Significant clustering of high transmission intensity subdistricts was observed in central-southeast Jakarta, predominantly in the region of East Jakarta, while spatial clustering of low transmission intensity subdistricts was observed in the region of Central Jakarta (Fig 3 and S5 Fig in S1 Text). Using force of infection estimates from model 1 variant *PS* fitted to the data averaged over the period 2008–2017, we observed a weak correlation between subdistrict force of infection and population density, with a 0.063 (95%CI: -0.009–0.136) average increase in force of infection per 100,000 unit increase in population density. Population density explained approximately

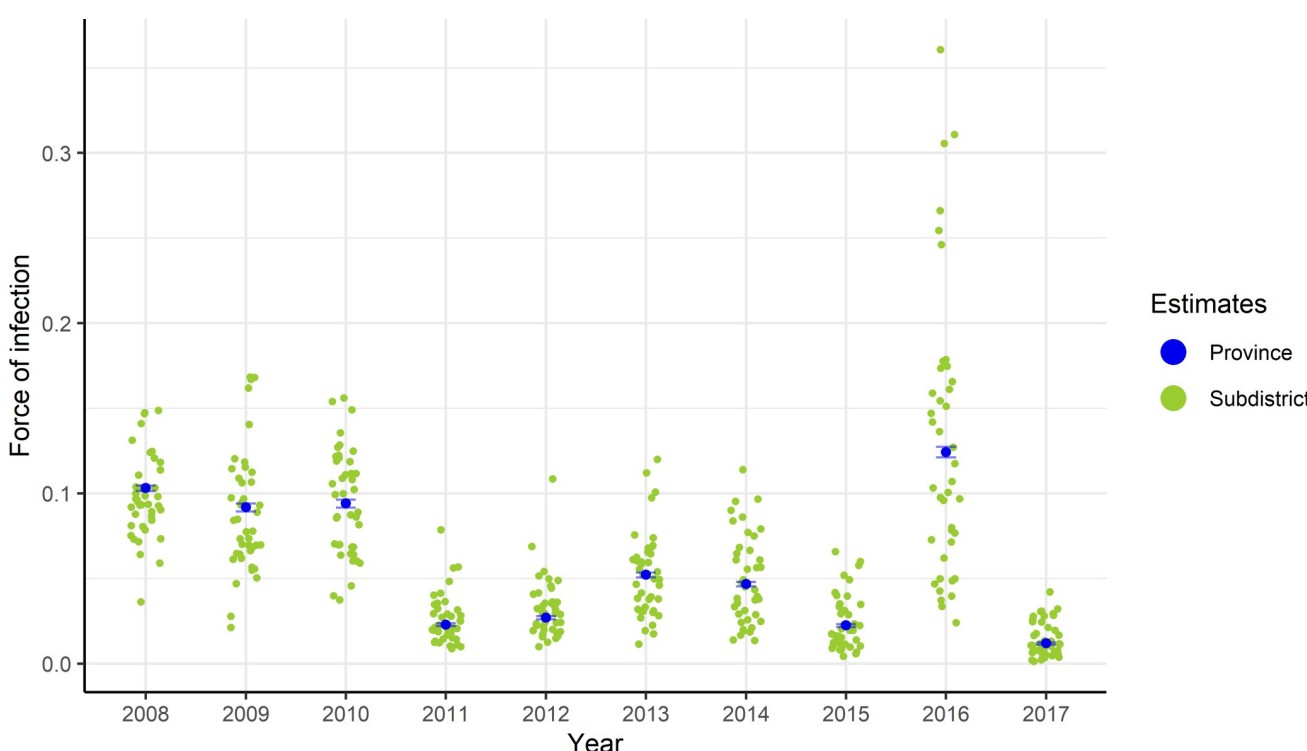

**Fig 2. Total force of infection (4λ) estimates obtained from the fit of model 2 variant PS (assuming reported hospitalised DHF cases are caused by primary or secondary infections) to annual DHF cases reported between 2008 and 2017.** Blue points show the median and the bars the 95% credible interval of the estimates obtained for the whole province of Jakarta. Green points represent the yearly median estimates obtained at the subdistrict level. Annual subdistrict parameter estimates are available in S2 File.

5% (adjusted $R^2$ = 0.049) of the variability in annual average subdistrict force of infection estimates (S8 Fig in S1 Text).

Using the average annual force of infection estimates obtained with model 1 variant *PS* from the 10-year period, we derived the expected proportion of seropositive population by age in Jakarta province and estimated the extent to which this proportion would vary between the highest and lowest transmission intensity subdistricts within Jakarta, as shown in Fig 4A. All subdistrict estimates of the proportion of seropositive 9-year olds are given in Table E in S1 Text. At age 9, the expected proportion of seropositive individuals in Jakarta province is 68.9% (95%CrI: 68.6–69.3) ranging from 55.6% (95%CrI: 50.2–60.4) to 77.1% (95%CrI: 74.8–79.1) across subdistricts. By age 15, this proportion is expected to have risen to 85.8% (95%CrI: 85.5–86.0) for the entire province, ranging from 74.2% (95%CrI: 68.7–78.7) to 91.4% (95%CrI: 89.9–92.7) across subdistricts. Fig 4B shows the expected proportion of subdistricts that reach 50%, 70% and 90% seroprevalence by age. We find that all subdistricts (42/42) in Jakarta are expected to have at least 50% seroprevalence amongst 9-year olds, and 38% (16/42) are expected to have seroprevalence levels above 70% at this age. At age 15, all subdistricts (42/42) are expected to have seroprevalence levels of at least 70%, with 12% (5/42) of subdistricts expected to have seroprevalence levels of 90% or more in this age group.

## Discussion

We found that the force of infection estimate obtained with model 1 variant *PS* fitted to average annual case notification data reported between 2008 and 2017 in Jakarta province (0.130 (95%CrI: 0.129–0.131)) was similar to the estimate obtained from seroprevalence data

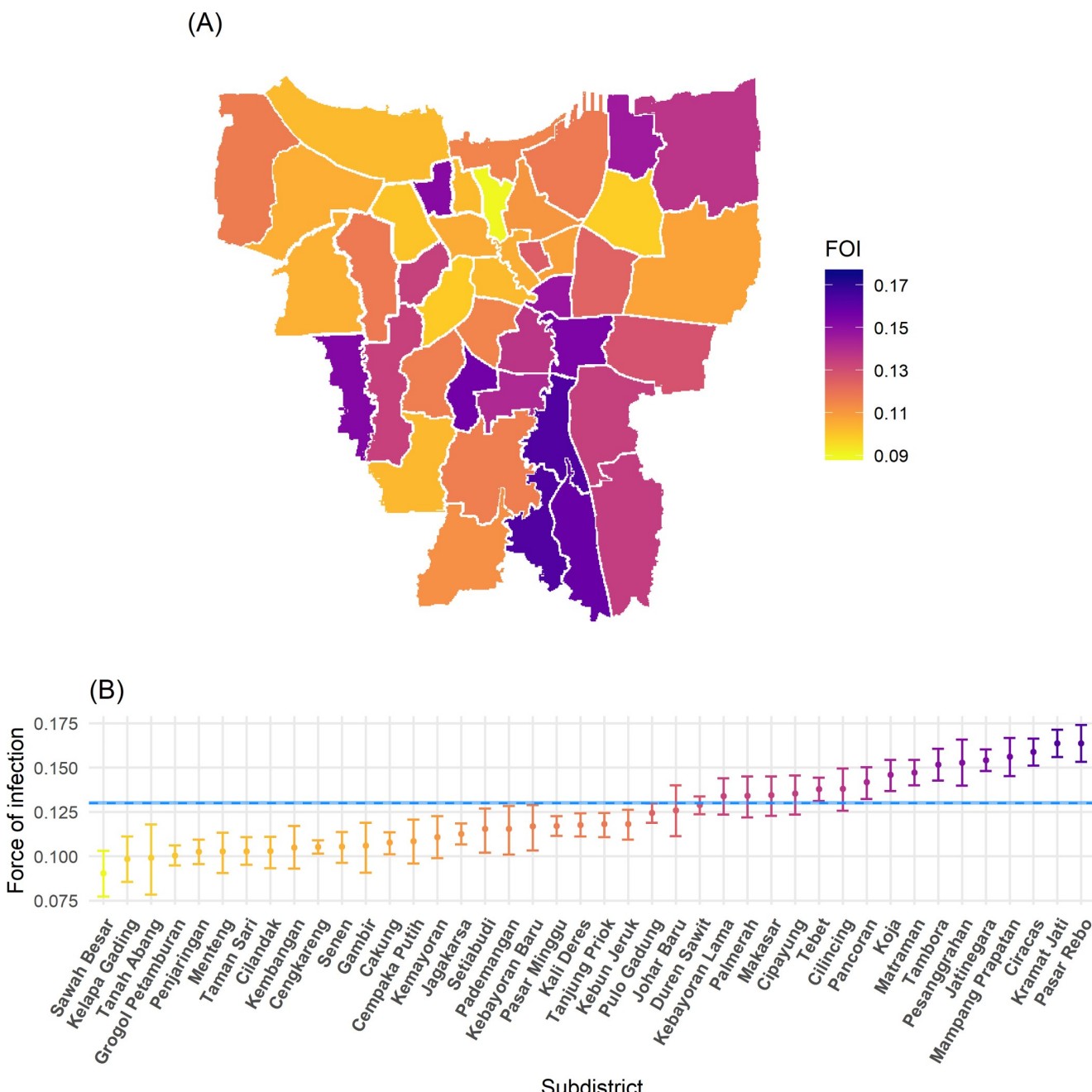

**Fig 3. Spatial variation in average annual total dengue force of infection (4λ) estimates in Jakarta obtained from the fit of model 1 variant PS (assuming reported hospitalised DHF cases are caused by primary or secondary infections) to average annual hospitalised DHF data reported in 2008–2017.** (A) Map of median force of infection estimates per subdistrict; (B) median and 95% credible interval of the estimates reported in panel A. Blue dashed line and shading shows the median and 95% credible interval province-level average annual force of infection in 2008–2017. FOI: force of infection. Boundaries were obtained from Wikimedia Commons under a CC-BY 3.0 NL license and converted to shapefile format using QGIS (25,26).

collected in 2014 in 30 urban subdistricts of Indonesia (0.14 (95%CI: 0.133–0.147)) [5,25]. We observed significant spatiotemporal heterogeneity in force of infection within Jakarta province, with long-term spatial clustering of both high and low transmission intensity subdistricts. Though ranking of subdistrict transmission intensity estimates varied by year, our analysis

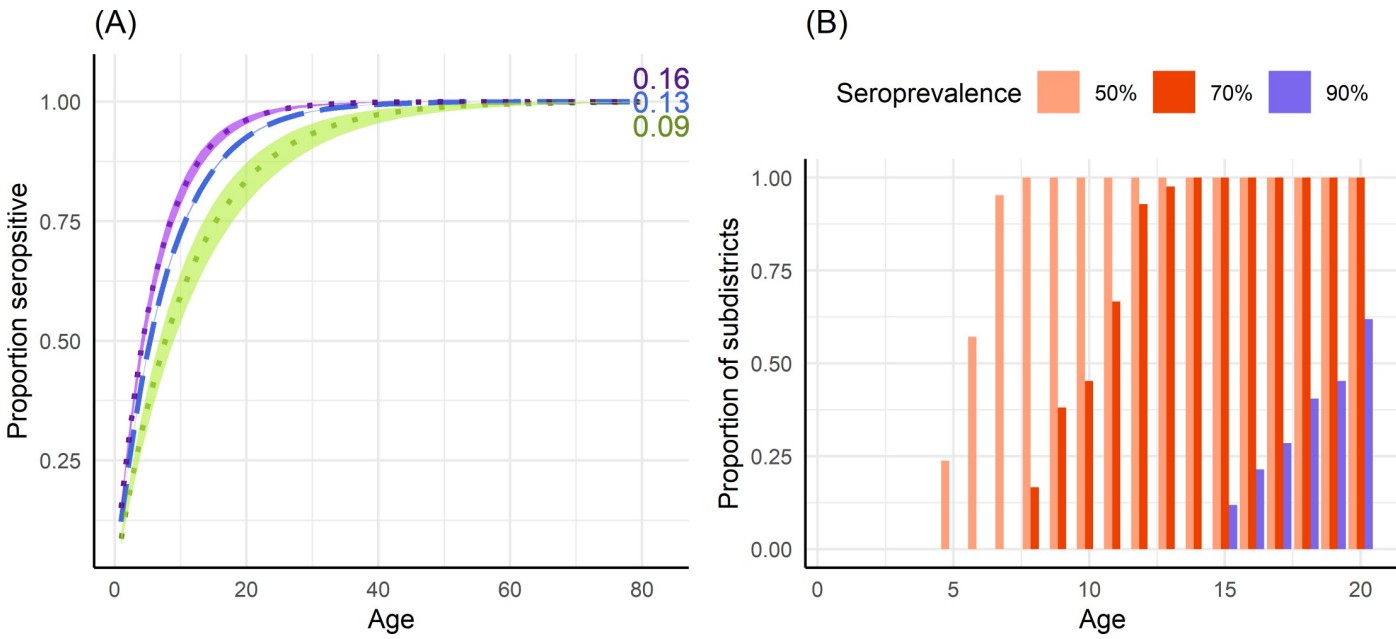

**Fig 4. Expected seroprevalence levels in Jakarta province.** (A) Expected proportion (median and 95% credible interval) of the population seropositive to dengue by age using the total force of infection (4λ) estimate obtained from the fit of model 1 variant PS (assuming all hospitalised DHF cases are caused by primary or secondary infections) to average annual hospitalised DHF data reported in the period 2008–2017 in Jakarta province (blue dashed line); Sawah Besar, the subdistrict with the lowest force of infection estimate (green); and Pasar Rebo, the subdistrict with the highest force of infection estimate (purple). The corresponding average force of infection estimate is given at the end of each line. (B) The proportion of subdistricts (N = 42) in Jakarta province that reach at least 50%, 70% and 90% seroprevalence by age, obtained from subdistrict force of infection estimates using model 1 variant PS fitted to the average annual incidence of hospitalised DHF cases reported in the period 2008–2017.

identified a long-term average hot-spot of dengue transmission in the southeast of Jakarta province and clustering of low transmission intensity in the region of Central Jakarta. Subdistrict population density was found to have a weak association with dengue transmission intensity, explaining approximately 5% of the variation in subdistrict force of infection estimates during 2008–2017, with higher average annual force of infection associated with greater population densities (for details see S1 Text sections 3.5 and S8 Fig). Though this association may suggest higher rates of human-vector contact in densely populated areas, numerous other environmental, socioeconomic and behavioural factors not included in this analysis are likely to influence local transmission intensity.

Annual reporting probabilities varied by year, from 0.024 (95%CrI: 0.022–0.026) in 2017 to 0.179 (95%CrI: 0.176–0.182) in 2016, the year in which a significant outbreak took place. Similar estimates have been observed in a prospective cohort study of dengue infections in Thai children aged 4–16, where 12.5% of secondary infections resulted in hospitalized dengue cases [26]. Annual transmission intensity estimates for Jakarta province during 2008–2017 ranged from 0.012 (95%CrI: 0.011–0.013) in 2017 to 0.124 (95%CrI: 0.121–0.128) in 2016, with large heterogeneity at the subdistrict level. This temporal variability in annual dengue force of infection means that the expected proportion of dengue-naïve children in any one age group also varies by year which, coupled with complex immunological factors, such as antibody-dependent enhancement, will shape the risk of severe dengue in children in any given year. Results from recent analyses of a longitudinal study conducted in Thailand found a significant cohort effect on the proportion of 9-year olds who were dengue-naïve by study year, with up to a two-fold difference in the probability of being dengue-naïve depending on year [27]. When assessing the level of dengue transmission from case notification data, it is therefore important to

consider data from multiple years in order to obtain reliable long-term estimates of average transmission intensity which will hold relevance for public health policy decision-makers.

Recent studies [28,29] have demonstrated that vaccination campaigns with the CYD-TDV vaccine could potentially increase individual-level risks of disease in low-to-medium transmission intensity settings, highlighting the importance of tailoring control and preventative programmes to the local setting. While the World Health Organization (WHO) initially advised that countries with geographic settings reaching at least 70% seroprevalence by 9 years of age could consider CYD-TDV vaccination [30], new data from Sanofi Pasteur [31,32] produced a change in the vaccine recommendations, with CYD-TDV vaccination currently advised only among subjects ≥9 years of age with evidence of a previous dengue infection [9,33]. Though temporal variability in force of infection means that the proportion of seropositive individuals in any one age group will also vary by year, our estimates from data averaged across the 10-year period represent long-term averages in transmission intensity. Using long-term average transmission intensity estimates, we expect average seroprevalence levels of 59.1% (95% CrI: 58.7–59.4), 78.6% (95%CrI: 78.3–78.9) and 88.8% (95%CrI: 88.6–89.0), in the 5–9, 10–14 and 15–19 age groups, respectively, in Jakarta province. Additionally, we expect every subdistrict in Jakarta to have seroprevalence levels of at least 50% amongst 9-year olds, suggesting that across all subdistricts, at least one in every two 9-year olds that are serologically screened would be eligible for vaccination. The age-specific seroprevalence estimates presented in this study can inform potential future individual-based serological screening programmes to determine the proportion of screened individuals that would be eligible for vaccination.

Overall, we find that the $S$ and $PS$ model variants produce largely similar estimates of average force of infection, except for years when substantial proportions of DHF cases are estimated to be caused by primary infections ($\gamma_1$). For these years, as expected, model variant $S$ estimates a higher force of infection than model variant $PS$, due to the assumption that all DHF cases have experienced two infections. In both model variants $S$ and $PS$, we assumed that post-secondary infections do not cause severe disease and are hence not detected. Though post-secondary infections are thought to be largely asymptomatic due to long-term heterotypic immunity, if a considerable proportion of severe dengue cases were being caused by tertiary or quaternary infections (perhaps in older individuals whose cross-protective immunity had waned) our models could potentially underestimate the force of infection. As shown in [14], a low age of seroconversion and a high age of DHF are easier to reconcile when relaxing the assumption of clinical protection after secondary infection, which is important to consider given the high occurrence of post-secondary infections observed in hyperendemic regions of Indonesia [34] as well as the increasing average age of DHF cases that has been observed in the country [35].

There are potential limitations in the assumptions adopted in this analysis which are important to consider when interpreting the estimates obtained with our models. When estimating transmission intensity within administratively-defined boundaries, we inherently assumed that cases reported in any one administrative unit were also infected in that same administrative unit, which is less realistic at small administrative levels such as subdistricts. While we explicitly accounted for temporal trends, we assumed no spatial interactions among Jakarta subdistricts. Accounting for spatial structure would increase the complexity of the inferential problem and make parameter inference challenging. Furthermore, we assumed reporting probabilities to be constant in time and age, though fluctuations in reporting may occur over time due to situational awareness (e.g. during epidemics) or by age if any age-related reporting biases were occurring. Additionally, the parameter results shown in this analysis are estimated assuming no overdispersion in the distribution of dengue cases (i.e. we assume a Poisson distribution for the total number of cases and a multinomial distribution for the age distribution

of cases), which may result in an underestimation of the uncertainty around the central parameter estimates. In exploratory analysis, we tested the use of over-dispersed case distributions (specifically assuming a negative binomial distribution for the total number of cases and a Dirichlet multinomial likelihood for the age-distribution of cases). However, the use of over-dispersed distributions hindered MCMC convergence. A significant dengue outbreak occurred in Jakarta in 2016, for which our model could not entirely reproduce the high rates of disease observed in the 10–14 years age group (S2 Fig in the S1 Text). This is likely due to our assumption of equal transmissibility of all 4 serotypes, which does not account for varying levels of population immunity to different serotypes. It is probable that the epidemic in 2016 was the result of the emergence of a newly dominant serotype to which the population had low levels of prior immunity. Due to a lack of serotype-specific data, this is not accounted for in our model, potentially resulting in an underestimation of the force of infection in 2016. In addition, dengue transmission in Jakarta has a distinct seasonal pattern, with case numbers typically peaking in March through May, following the rainy season [36]. In this analysis we assumed that the force of infection is constant within each year, ignoring intra-annual variability and thus producing average annual transmission intensity estimates. Long-term estimates of transmission intensity, however, will hold relevance to decisions when considering the implementation of future control and prevention programmes.

In conclusion, using publicly available data on DHF cases reported in Jakarta in the 10-year period of 2008–2017, we have presented the first detailed analysis of the spatiotemporal variation in dengue transmission intensity conducted to date in Jakarta province. This study highlights the importance of estimating dengue transmission intensity from the analysis of age-stratified dengue case notification data rather than inferring it from the absolute number of cases, which can often be misleading. The estimates presented in this study will provide invaluable insights into the potential impact and cost-effectiveness of future control and preventative programmes, including vaccination with current or future dengue vaccines, and the use of the *Wolbachia* technology, in the province of Jakarta.

## Supporting information

**S1 Text. Supplementary information.**
(DOCX)

**S1 File. Subdistrict population age-structures by year.**
(CSV)

**S2 File. Annual subdistrict parameter estimates.**
(CSV)

## Author Contributions

**Conceptualization:** Megan O'Driscoll, Natsuko Imai, Neil M. Ferguson, Ilaria Dorigatti.

**Formal analysis:** Megan O'Driscoll.

**Methodology:** Megan O'Driscoll, Natsuko Imai, Neil M. Ferguson, Ilaria Dorigatti.

**Supervision:** Natsuko Imai, Neil M. Ferguson, Ilaria Dorigatti.

**Visualization:** Megan O'Driscoll.

**Writing – original draft:** Megan O'Driscoll.

**Writing – review & editing:** Megan O'Driscoll, Natsuko Imai, Neil M. Ferguson, Sri Rezeki Hadinegoro, Hindra Irawan Satari, Clarence C. Tam, Ilaria Dorigatti.

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
