## [Decision Letter · Decision Letter 0]

28 Nov 2019

Dear Ms O'Driscoll:

Thank you very much for submitting your manuscript "Spatiotemporal Variability in Dengue Transmission Intensity in Jakarta, Indonesia" (#PNTD-D-19-01147) for review by PLOS Neglected Tropical Diseases. Your manuscript was fully evaluated at the editorial level and by independent peer reviewers. The reviewers appreciated the attention to an important problem, but raised some substantial concerns about the manuscript as it currently stands. These issues must be addressed before we would be willing to consider a revised version of your study. We cannot, of course, promise publication at that time.

We therefore ask you to modify the manuscript according to the review recommendations before we can consider your manuscript for acceptance. Your revisions should address the specific points made by each reviewer. 

When you are ready to resubmit, please be prepared to upload the following:

(1) A letter containing a detailed list of your responses to the review comments and a description of the changes you have made in the manuscript.

(2) Two versions of the manuscript: one with either highlights or tracked changes denoting where the text has been changed (uploaded as a "Revised Article with Changes Highlighted" file); the other a clean version (uploaded as the article file).

(3) If available, a striking still image (a new image if one is available or an existing one from within your manuscript). If your manuscript is accepted for publication, this image may be featured on our website. Images should ideally be high resolution, eye-catching, single panel images; where one is available, please use 'add file' at the time of resubmission and select 'striking image' as the file type. 

Please provide a short caption, including credits, uploaded as a separate "Other" file. If your image is from someone other than yourself, please ensure that the artist has read and agreed to the terms and conditions of the Creative Commons Attribution License at http://journals.plos.org/plosntds/s/content-license (NOTE: we cannot publish copyrighted images). 

(4) If applicable, we encourage you to add a list of accession numbers/ID numbers for genes and proteins mentioned in the text (these should be listed as a paragraph at the end of the manuscript). You can supply accession numbers for any database, so long as the database is publicly accessible and stable. Examples include LocusLink and SwissProt.

(5) To enhance the reproducibility of your results, we recommend that you deposit your laboratory protocols in protocols.io, where a protocol can be assigned its own identifier (DOI) such that it can be cited independently in the future. For instructions see http://journals.plos.org/plosntds/s/submission-guidelines#loc-methods

While revising your submission, please upload your figure files to the Preflight Analysis and Conversion Engine (PACE) digital diagnostic tool, https://pacev2.apexcovantage.com/ PACE helps ensure that figures meet PLOS requirements. To use PACE, you must first register as a user. Then, login and navigate to the UPLOAD tab, where you will find detailed instructions on how to use the tool. If you encounter any issues or have any questions when using PACE, please email us at figures@plos.org.

We hope to receive your revised manuscript by Jan 27 2020 11:59PM. If you anticipate any delay in its return, we ask that you let us know the expected resubmission date by replying to this email.

To submit a revision, go to https://www.editorialmanager.com/pntd/ and log in as an Author. You will see a menu item call Submission Needing Revision. You will find your submission record there. 

Sincerely,

Andrew S. Azman

Deputy Editor

Reviewer's Responses to Questions

**Key Review Criteria Required for Acceptance?**

**Methods**

-Are the objectives of the study clearly articulated with a clear testable hypothesis stated?

-Is the study design appropriate to address the stated objectives?

-Is the population clearly described and appropriate for the hypothesis being tested?

-Is the sample size sufficient to ensure adequate power to address the hypothesis being tested?

-Were correct statistical analysis used to support conclusions?

-Are there concerns about ethical or regulatory requirements being met?

Reviewer #1: (No Response)

Reviewer #2: Although I think the article has merit as a showcase of the application to Indonesian dengue control program, methodological concerns need to be addressed to ensure validity of the analyses.

- I understand that there might be computing issues with complicated models. However, there are examples that changing model assumptions, i.e. over-dispersed distributions in this case, can alter results significantly. How would you justify your results if you cannot apply the over-dispersed models? 

- On page 10, it’s not clear from model 2 if the authors apply equations 3-6 independently for each subdistrict or as the whole province because the spatial index is not specified. It’s also evident from spatial association metrics that spatial correlation is present in the data but does not seem to be captured in model 2. 

- Line 254, the authors assumed uniform priors on all parameters. Please provide the range of the priors.

**Results**

-Does the analysis presented match the analysis plan?

-Are the results clearly and completely presented?

-Are the figures (Tables, Images) of sufficient quality for clarity?

Reviewer #1: (No Response)

Reviewer #2: (No Response)

**Conclusions**

-Are the conclusions supported by the data presented?

-Are the limitations of analysis clearly described?

-Do the authors discuss how these data can be helpful to advance our understanding of the topic under study?

-Is public health relevance addressed?

Reviewer #1: (No Response)

Reviewer #2: I like that the authors list out the study limitations. However, I wonder how realistic and useful this work would be for the dengue control program under those assumptions. For example, the authors allow for post-secondary infection to contribute to transmission which do not cause severe conditions. This is quite a strong assumption and difficult to justify. Since the authors are working with policy makers, that would be more useful to the control program to have results under different assumptions rather than just one instance presented to the government.

**Editorial and Data Presentation Modifications?**

Reviewer #1: (No Response)

Reviewer #2: (No Response)

**Summary and General Comments**

Reviewer #1: The authors present an approach for estimating the force of infection from age-stratified case reports. The manuscript is well written and the methods and results are clearly described. The findings are presented in light of dengue vaccination campaigns (among others), where estimates of the pre-exposed population at a certain age are important for the implementation of test-before-vaccination approaches. The manuscript is of good quality and I recommend it for publication. I only have a few minor questions and suggestions: 

1. The authors estimate reporting and misreporting rates. It would be worthwhile to report and discuss these in the manuscript. How to the estimated reporting rates differ between primary and secondary infections? How do estimates compare with published estimates (for instance from JD Stanaway et al, 2016, Lancet)?

2. The authors report hotspots of infection, i.e., areas where transmission is consistently higher than elsewhere. One might expect that, after a big outbreak year, the force of infection in such hotspots is lower due to build up immunity. Understanding such mechanisms could be helpful for targeting control efforts. Do the authors see any such pattern? Is the ranking across sub-districts stable over the years?

3. I am not sure all state variables are described in the text. (M is equation 4, maybe others). Please check if all are included.

Reviewer #2: (No Response)

PLOS authors have the option to publish the peer review history of their article (what does this mean?). If published, this will include your full peer review and any attached files.

Reviewer #1: No

Reviewer #2: No

---

## [Editor Report · Decision Letter 1]

29 Jan 2020

Dear Ms O'Driscoll,

We are pleased to inform you that your manuscript 'Spatiotemporal Variability in Dengue Transmission Intensity in Jakarta, Indonesia' has been provisionally accepted for publication in PLOS Neglected Tropical Diseases.

Before your manuscript can be formally accepted you will need to complete some formatting changes, which you will receive in a follow up email. **In addition, please make sure that you add a link to your code online and remove the track changes edits from the supplement in the next version. **A member of our team will be in touch within two working days with a set of requests.

Best regards,

Andrew S. Azman

Deputy Editor

Andrew Azman

Deputy Editor
